# Risk Factors from Pregnancy to Adulthood in Multiple Sclerosis Outcome

**DOI:** 10.3390/ijms23137080

**Published:** 2022-06-25

**Authors:** Enrique González-Madrid, Ma. Andreina Rangel-Ramírez, María José Mendoza-León, Oscar Álvarez-Mardones, Pablo A. González, Alexis M. Kalergis, Ma. Cecilia Opazo, Claudia A. Riedel

**Affiliations:** 1Laboratorio Endocrinología-Inmunología, Departamento de Ciencias Biológicas, Facultad de Ciencias de la Vida, Universidad Andrés Bello, Santiago 8320000, Chile; e.gonzlezmadrid@uandresbello.edu (E.G.-M.); m.rangelramirez@uandresbello.edu (M.A.R.-R.); m.mendozaleon@uandresbello.edu (M.J.M.-L.); o.alvarezmardones@uandresbello.edu (O.Á.-M.); 2Millennium Institute on Immunology and Immunotherapy, Santiago 8320000, Chile; pagonzalez@bio.puc.cl (P.A.G.); akalergis@bio.puc.cl (A.M.K.); cecilia.opazo@gmail.com (M.C.O.); 3Departamento de Genética Molecular y Microbiología, Facultad de Ciencias Biológicas, Pontificia Universidad Católica de Chile, Santiago 8320000, Chile; 4Departamento de Endocrinología, Escuela de Medicina, Facultad de Medicina, Pontificia Universidad Católica de Chile, Santiago 8320000, Chile; 5Instituto de Ciencias Naturales, Facultad de Medicina Veterinaria y Agronomía, Universidad de Las Américas, Manuel Montt 948, Providencia 7500000, Chile

**Keywords:** thyroid hormones, gestational period, hypothyroxinemia, immune response, risk factors, multiple sclerosis, experimental autoimmune encephalomyelitis

## Abstract

Multiple sclerosis (MS) is an autoimmune disease characterized by a robust inflammatory response against myelin sheath antigens, which causes astrocyte and microglial activation and demyelination of the central nervous system (CNS). Multiple genetic predispositions and environmental factors are known to influence the immune response in autoimmune diseases, such as MS, and in the experimental autoimmune encephalomyelitis (EAE) model. Although the predisposition to suffer from MS seems to be a multifactorial process, a highly sensitive period is pregnancy due to factors that alter the development and differentiation of the CNS and the immune system, which increases the offspring’s susceptibility to develop MS. In this regard, there is evidence that thyroid hormone deficiency during gestation, such as hypothyroidism or hypothyroxinemia, may increase susceptibility to autoimmune diseases such as MS. In this review, we discuss the relevance of the gestational period for the development of MS in adulthood.

## 1. Introduction

Multiple sclerosis (MS) is one of the most frequent autoimmune diseases worldwide that affects the central nervous system (CNS) and manifests in those affected as severe physical, cognitive, and neurological impairments [1,2,3,4]. According to current estimations, MS affects ~2.8 million people worldwide, most of whom are young adult women [5]. The autoimmune response in MS is mainly driven against self-antigens from the myelin sheath and neuronal cells [6,7,8]. It is characterized by immune cell infiltration into the CNS, specifically by Th1 and Th17 CD4^+^ T cells that promote local inflammation and disrupt immunological tolerance, thereby triggering oligodendrocyte and neuronal loss [6,7,8].

The causes of MS are diverse and multifactorial, and it may also originate in the gestational period. This review will discuss the gestational factors, such as maternal thyroid hormone deficiency, that could predispose the progeny to suffer from more intense MS in adulthood. This notion is based on the fetal programming concept, whereby epigenetic and permanent modifications are acquired during gestation, which could increase the susceptibility of the progeny to different autoimmune disorders, such as MS [8].

## 2. Multiple Sclerosis

Multiple sclerosis (MS) is an autoimmune disease developed mainly against CNS antigens, which triggers neurodegeneration [2,3,4]. Autoimmunity is raised against self-antigens from the myelin sheath and neuronal proteins, causing tissue damage that will affect synaptic transmission and will negatively impact nerves [2,3,4]. Clinical signs of MS are numbness or weakness of the limbs; electric shock sensations that produce involuntary movements and tremors accompanied by impaired coordination and unsteady walking; partial or complete loss of vision; dizziness; fatigue; and tingling of different body parts [3,9]. Symptom variability is associated with the spatio-temporal appearance of the injury sites within the CNS [10]. MS is characterized by confluent demyelinated areas, called plaques, in the brain and spinal cord’s white and gray matter with a loss of myelin sheaths and oligodendrocytes [10,11]. The ongoing disease results in gradual neuroaxonal function loss related to patient disabilities and brain atrophy accompanied by ventricular enlargement [10,11]. The course of MS is highly variable, but there are three main stages: (I) a pre-clinical stage detected only by MRI imaging in which the first brain lesions are observed; (II) the onset of recurring symptoms, followed by partial or total recovery known as relapsing–remitting multiple sclerosis (RRMS); and (III) a progressive stage in which substantial neuronal damage affects the patient’s motor capacity [9,12,13]. After approximately 15 years, the disease’s symptoms become permanent in 50% of untreated patients, causing prominent and progressive deterioration. This disease form is known as secondary progressive multiple sclerosis (SPMS) [9,13]. 

On the other hand, ~10–15% of MS cases gradually develop more intense neurological deterioration from the onset of MS, known as primary progressive MS (PPMS), which is more common in men than in women [12]. PPMS presents fewer lesions in the white matter with increased microglia activation and axonal loss leading to spinal cord atrophy [12]. Finally, a less common form of MS is the progressive-relapsing MS (PRMS), which is characterized by a constant worsening since the beginning of the disease, caused by extreme nerve damage and loss [2,12,13]. There are occasional relapse episodes with similar symptoms that intensify the RRMS, caused by inflammation in nerves and spinal cord [2,12,13]. According to the atlas of MS (third edition, 2020), it is estimated that the prevalence of MS is 35.9 per 100,000 individuals in the population worldwide; approximately 2.8 million people currently live with MS [5], and it is more frequent in young adult women than in men [5]. This number has increased since 2013, when there were 2.3 million people diagnosed with MS [14]. Nevertheless, the average age of disease onset has been maintained at 30 years old, and, approximately 25 years after diagnosis, ~50% of patients require permanent use of a wheelchair [5,14]. The global prevalence of MS increases with distance from the equator; it is more common in the north European population [5,13,14]. In this regard, San Marino (337 per 100,000), Germany (303 per 100,000), and Denmark (282 per 100,000) are the three countries that have the highest prevalence of MS in Europe and worldwide [5]. In North America, the United States has the highest informed prevalence in the region (288 per 100,000); however, the American continent has only 112 per 100,000. Interestingly, in North America, ethnicities such as Hispanic Americans and Afro-Americans show a faster disease progression than White Americans [15]. In the rest of the WHO regions, the reported MS prevalence decreases. This is the case for the eastern Mediterranean (30 per 100,000), followed by southeast Asia (9 per 100,000), and, finally, Africa and the western Pacific (both with 5 per 100,000) [5]. Treating comorbidities in MS patients without aggravating any secondary condition represents a medical challenge [16]. Although the prevalence of MS comorbidities is high in North America and Europe, little is known about MS comorbidities in South and Central America, Africa, and Asia [16]. The most frequent comorbidities observed in these patients are asthma, anxiety, depression, diabetes, hypertension, hyperlipidemia, vascular disease, cardiac arrhythmias, ischemic heart disease, chronic pulmonary disease, and psychosis [16,17].

## 3. Interplay between the Adaptative and Innate Immune Response in MS

The characteristic lesions of MS are produced by inflammation, demyelination, oligodendrocyte loss, gliosis, and neuroaxonal degeneration, promoted by immune cell infiltration across the blood-brain barrier (BBB) [6]. Imaging and pathological studies point toward the adaptative immune cells as essential players in MS pathogenesis [18]; however, the innate immune response also plays an essential role in MS. The innate immune system is the first defense against pathogens and foreign molecules, but it has also been related to autoimmune disorders [19]. The system recognizes conserved pathogen-associated molecular patterns (PAMPs) derived from microbes via the toll-like receptors (TLR) expressed in dendritic cells (DCs), macrophages, neutrophils, mast cells, lymphocytes, endothelial cells, and epithelial cells [19,20]. The recognition of PAMPs leads to activation of inflammatory responses, such as cytokine production and effector mechanisms, such as lysozyme production, phagocytosis, and ROS production [19,20]. In MS, innate immune cells are relevant for disease initiation and progression. For instance, activated macrophages induce a pro-inflammatory response of T and B cells and early microglial activation, one of the first events underlying MS lesions [21]. Regarding mast cells, which are predominant in allergic reactions, they are present in a low proportion in the CNS. Nevertheless, a relevant role in MS progression has been proposed due to a RANTES secretion, a mast cell chemoattractant that is increased in MS lesions [22,23]. Other mediators released by mast cells, such as histamine and tryptase, have also been found in higher concentrations in the CSF of MS patients [22,24]. These mediators increase leukocyte infiltration during the BBB aperture, improving their adhesion, rolling, and extravasation into the CNS compartment [22,23,24]. Furthermore, mast cells have been proposed to act as APCs due to their influence on Th1 responses in the context of MS; however, this notion has not yet been fully demonstrated [25]. In vitro analyses have shown that mast cell degranulation in response to MBP stimulation leads to demyelination and the destruction of oligodendrocytes and neurons [25]. On the other hand, phagocytic cells can generate high levels of ROS and RNS species that act as antimicrobial agents [25]. In MS lesions, the inducible nitric oxide synthase (iNOS) enzyme generates large amounts of nitric oxide (NO), inducing microglial and neuronal death. NO mediates the microglial cells’ cytotoxicity and oligodendrocytes’ necrosis in mice suffering from EAE [25]. Finally, natural killer (NK) cells and natural killer T cells (NKT) are decreased in MS patients [26]. NKT cells express invariant TCR α-chains, which are thought to link innate and adaptative responses. The reduction in NK and NKT cells contributes to the disease progression [26]. Altogether, these observations show an essential role for innate immune responses in MS pathogenesis. In this context, effector T cells and B cells are specifically recruited, probably by autoantigens against the myelin basic protein (MBP) in the CNS and released into the circulation [27]. Immune responses generated by these cells involve the expansion of antigen-specific lymphocytes from precursor cells in the lymph nodes, which were initially activated by antigen-presenting cells (APCs), such as professional dendritic cells (DCs) [27]. T cell organization within the lesions has been described as the “outside-in” hypothesis. T CD8^+^ lymphocytes present a periphery localization, whereas T CD4^+^ lymphocytes present a centric localization within the lesions [27,28]. Once in the CNS, autoreactive T lymphocytes are reactivated by APCs at the CNS parenchyma, recruiting more T lymphocytes and innate immune cells, and generating inflammatory lesions [29]. This immune interplay is summarized in (Figure 1). Within the subtypes of T CD4^+^ lymphocytes related to MS, IFN-γ and TNF-α-producing Th1 lymphocytes, and IL-17-producing CD4^+^ T lymphocytes (Th17) cells, which have a pivotal role in disease progression, can be found in MS lesions [29]. Th17 cells also secrete IL-21, IL-22, and TNF-α. These cytokines induce B cell activation and Th1 lymphocyte polarization, which promote inflammation [29].

Analysis of cortical lesions, perivascular CNS, and cerebrospinal fluid (CSF) showed the presence of IL-17, IFN-γ, and cytolytic granules produced by cytotoxic CD8^+^ T cells, suggesting a role for these cells in MS pathogenesis [28,29]. CD8^+^ T cells ubiquitously express the major histocompatibility complex class I (MHC-I), which intriguingly is also expressed in the oligodendrocytes, astrocytes, and neurons of MS patients, suggesting that these antigen-presenting glial cells may increase the frequency of pathogenic CD8^+^ T cells through antigen presentation to reactive T cells in the brain parenchyma [30]. These reactive cytotoxic cells also secrete IFN-γ and induce cell death of myelin-expressing cells, favoring disease progression [30]. Immunohistochemical analyses of postmortem brains showed the presence of cytotoxic cytokines and lytic granules found primarily in projection neurons located in the outermost layers of the cortex, which are essential for linking remote cortical regions into a networked whole [30,31]. The presence of pathogenic CD8^+^ T cells induces a pro-inflammatory environment, leading to the activation of immune resident cells, such as microglia, astrocytes, and innate immune cells, and increasing the production of reactive oxygen species (ROS) and reactive nitrogen species (RNS) [32]. Altogether, these events contribute to myelin loss, oligodendrocyte destruction, and axonal damage, which leads to neuronal dysfunction [30,31,32]. 

Consistently, inflammation impairs the tolerogenic function of immunomodulatory cells by reducing expression of the anti-inflammatory cytokines favoring the disease course [33]. In this context, regulatory T cells (Tregs), which maintain immune tolerance by secreting IL-10 and TGF-β, have a relevant role in MS pathogenesis [33]. Two types of Treg cells have been identified in this disease: (i) FOXP3-expressing Tregs, which suppress effector T cell proliferation by cell–cell interactions; and (ii) the T regulatory cell subtype 1 (Tr1), which secretes anti-inflammatory cytokines (mainly IL-10) to regulate the immune response [33,34]. In MS patients, a decrease in their suppression capacity has been observed; however, no differences in cell proportions were found compared to healthy people [33,34]. These observations support the idea of an inflammatory T cell phenotype development or Treg function loss during MS, generating an immune environment that contributes to disease progression. Furthermore, B cells have also been related to MS pathogenesis as antigen presenters to brain T cells, activating them and inducing proliferation, pro-inflammatory cytokine, and chemokine production, as well as secretion of soluble factors that contribute to oligodendrocytes and neuronal damage [35]. The use of anti-CD20 therapy has evidenced a role for B cells since this antigen is expressed in these cells [36]. In fact, anti-CD20 therapy on MS patients has shown promising results in reducing the pro-inflammatory responses of autoreactive CD4^+^ and CD8^+^ T lymphocytes, limiting disease remission [36]. Peripheral B cells from MS patients secrete pro-inflammatory cytokines, particularly TNF-α, IL-6, and granulocyte–macrophage colony-stimulating factor (GM-CSF), which are thought to contribute to the increased polarization of naive T cells towards pro-inflammatory lineages, thereby accelerating MS progression [35,37]. On the other hand, a significant proportion of B cells can secrete anti-inflammatory cytokines, such as TGF-β and IL-10, which are known as regulatory B cells (Bregs). Therefore, they suppress cytokine production by dendritic cells and indirectly inhibit Th1 and Th17 differentiation [37]. Mice that are lacking B cells and induced with EAE are unable to recover from the condition, implying that Bregs provide valuable support to immunoregulatory function during MS pathogenesis [38]. This fact acquires relevance since it has been seen that the transfer of Breg cells into mice suffering from EAE allows their accumulation in lymph nodes and the spleen, where they drive the in vivo expansion of naïve T cells to Tr1 and Tregs phenotypes [39]. The number of Breg cells, defined as the CD19^+^CD25^+^ subset that is present in humans diagnosed with MS, is decreased [40], unaltered [41], or even increased [42]. Therefore, there is no clear consensus; nonetheless, the Breg cells’ functions are impaired in MS patients because they are deficient in secreting IL-10 [37,42]. Under normal conditions, B cells do not cross the BBB. However, they infiltrate the CNS, due to the altered BBB permeability observed in inflammatory diseases such as MS [43]. In MS patients, the B cells and antibody content are increased in the CSF but not in the serum, suggesting local production of antibodies [44]. B cells and antibodies contribute to disease progression in two ways: first, they can function as APCs and/or co-stimulators of autoreactive T cells through CD40/CD40-L signaling and, second, as producers of myelin-specific antibodies that could activate these autoreactive T cells. Furthermore, antibodies can promote demyelination by opsonizing myelin and consequent phagocytosis [45]. Several studies have focused on autoantibodies against the MBP and other CNS components. Using an MBP/PLP fusion peptide to induce EAE in C57BL/6 mice has shown that EAE’s disease progression is an antibody-dependent process [46]. Moreover, B cell-deficient mice induced with MBP/PLP-specific antibodies, followed by immunization with the fusion protein MP, showed a disease progression and severity similar to those seen in *wild-type* control mice [47]. Finally, other immunoglobulins (Igs) against other myelin components have been found in MS patients [48]. The myelin oligodendrocyte glycoprotein (MOG) is the most studied candidate as a B cell–autoantigen in MS because anti-MOG antibodies can also induce myelin destruction in the EAE model [48,49]. Indeed, the B cells’ responses to auto-MOG antibodies are enhanced in MS [49]. Anti-MOG antibodies are found in MS patient lesions and serum samples; however, they are also frequent in healthy people [48,49]. The role of B cells requires more research, as there are still many questions to be answered.

## 4. Genetic Susceptibility

Epidemiological studies in Europe and North America have shown that 7.3% of the population may present a genetic susceptibility to MS development, with women being more susceptible than men [50]. Research based on single nucleotide polymorphisms (SNPs) that use a large-scale genome-wide association (GWAS) has revealed at least 200 SNPs in the innate and adaptive immune cells of MS patients [51,52,53,54]. In addition, the genetic variants of vitamin D’s metabolism have also been associated with MS [55,56,57], a topic that will be addressed later in this review. 

Interestingly, there is a multifaceted association between obesity and MS. High body mass index (BMI) genetic variants in adolescent and adult patients are associated with the onset of MS [58,59]. Autoimmune diseases are usually associated with polymorphisms of the genes that regulate the function of the immune system; in this context, the HLA antigen D-related β1 subunit (DRB1) gene (HLA DRB1*15:01), within the major histocompatibility complex, has been identified as the primary locus associated with this correlation [58,60]. Furthermore, obesity is also correlated with the development of MS due to the polymorphisms found in fat mass obesity (FTO)-related genes [61,62]. The most attractive candidate is the FTO rs9939609 polymorphism in the α-ketoglutarate-dependent dioxygenase, an enzyme highly expressed in the nucleus of the cell. Despite the function of the α-ketoglutarate-dependent dioxygenase being unknown, it has been proposed to regulate fat mass, adipogenesis, and energy homeostasis [62,63]. The FTO rs9939609 polymorphism has consistently been associated with a higher BMI across different populations, including the Kuwaiti population [63]. Another link between MS and obesity is the adipokine’s capacity to modulate the immune system by increasing pro-inflammatory response [58,59]. It is still necessary to deeply understand how these risk genetic variants are regulated and the pathways in which they are involved in the immune system of these patients [54]. 

Tobacco smoking has been described as an environmental factor that increases the susceptibility to MS development in patients with genetic risk. It has been observed that patients who smoke tobacco and have certain genetic risk factors, such as the absence of HLA-DRB5*01 (DR2a) and HLA-DRB1*15 (or DR2b) (two alleles that take part in DR15 haplotype), produce auto-reactive T cells in the lungs with an altered peptide antigen presentation capacity that promotes CNS-directed effector immunity [64,65]. This immune response can increase the CD4^+^ T cells’ autoreactivity in MS [66,67]. HLA-DR physiopathology is still under research, and it has been proposed that the occurrence of self-reactivity or “auto proliferation” in the pathogenesis of MS is linked to T cells and B cells, in which memory B cells can be upregulated and may express antigens in the brain that can be recognized by CD4^+^ T cells [68]. The regulation of mitosis and apoptosis in immune cells can be mediated by cell cycle-regulated antiapoptotic proteins, such as survivin, which is a protein encoded by the *Baculoviral IAP repeat-containing 5 (BIRC5)* gene, and, in RRMS patients, it has been found that the genetic polymorphisms located in this gene promote increased levels of survivin [69,70], which suggests a critical participation in the persistence of the inflammatory condition in MS patients by contributing to reduced self-tolerance [71,72]. Thus, survivin has been considered as a target for immunotherapeutic strategies [73]. In MS patients, studies performed on innate and adaptive cells, such as natural killer (NK) and microglia, a resident immune cell in the brain, have key roles in inducing T cell autoreactivity, with at least 200 autosomal susceptibility variants outside the major histocompatibility complex (MHC) locus and 552 susceptibility-related putative genes distributed in the cellular components in these cells [68]. Clinical studies have also shown the genetic component of the inflammatory process controlled by a high molecular weight complex known as the inflammasome, which is composed of innate immune receptors, such as the nod-like receptor protein 3 (NLRP3) that can be activated in response to molecular PAMPS or DAMPS. Dysregulated inflammasome activity has been linked to uncontrolled inflammation; in fact, a gain of the function variants of NLRP3 in the MS peripheral blood monocytes of MS patients can represent a risk related to MS severity, due to significantly increased production of the proinflammatory cytokine IL-1β [74,75,76].

## 5. Epigenetics Associated with MS Susceptibility

Epigenetic modifications, including DNA methylation, histone modifications, and micro-RNA alterations, have been linked to the altered gene expression in MS pathogenesis, as observed in post-mortem brains and in animal models [77,78,79]. For instance, the increased expression of DNA methyltransferases DNMT1 (known as maintenance methyltransferase) and DNMT3A/3B (known as de novo methyltransferases) in the demyelinated hippocampi of MS brains suggests a role in the performance of brain cells for these enzymes [78]. Preferred DNA sites for methylation are CpG islands [80]. Interestingly, hypermethylation of the CpG sites at risk-genes’ promotors, such as *HLADRB1*, *IL2RA*, *IL7*, *IL7R*, *IL12RB1*, *IL12A*, *TNFRSF1A*, and *STAT3*, has been related to MS pathogenesis [77,81,82]. Epigenetic modifications analyses in CD4^+^ and CD8^+^ T cells from RRMS female patients present differentially methylated regions (DMR), with a hypermethylation pattern in HLA-DRB1, an independent MHC locus, RNF39 [83]. This pattern would be implicated in the predisposition of CD4^+^ T cells to become autoreactive; nevertheless, more studies are needed to dissect whether methylation at MHC is related to MS risk and its onset, or whether it is part of an altered methylation profile in MS patients [83]. This evidence is supported by epigenome-wide studies (EWAS), which have assessed changes in the whole DNA methylation patterns from peripheral blood mononuclear cells (PBMCs) and CD4^+^ and CD8^+^ T cells in MS patients [84,85]. Increased methylated CpG in the HLA locus of CD4^+^ T cells has been observed compared to control healthy individuals [84]. Moreover, hypermethylation of the CD8^+^ T cells has shown contrasting results [86]. Furthermore, Maltby and coworkers identified DMR at the lymphotoxin α (LTA) locus in CD19^+^ B cells from RRMS patients, which is thought to be related to the implication of these cells in MS pathogenesis [87]. Likewise, they found smaller DMRs in four well-known MS-associated genes: *SLC44A2*, *LTBR*, *CARD11*, and *CXCR5* [87]. Therefore, this evidence supports the notion that DNA methylation patterns are present in the MS context, which provides new perspectives for therapeutic interventions [81]. On the other hand, histone modifications also contribute to MS pathology [77,88]. The most common alterations of histones are phosphorylation, methylation, ubiquitylation, acetylation, deamination, and ADP ribosylation [89]. These post-translational modifications (PTMs) on histone proteins can alter the chromatin structure, thus altering the gene expression [90]. More specifically, acetylation and deacetylation modifications have been related to MS [77,88]. An increase in the expression of the histone acetylase enzymes and the acetylation of histone H3 has been found in chronic lesions of patients with MS and cultures of oligodendrocytes obtained from patients [91,92]. This epigenetic change is associated with an increased expression of the transcriptional inhibitors of oligodendrocyte differentiation, such as *nogo-A*, *LiNGo-1*, *semaphorin*, *ephrin*, and *netrin-1*, leading to impaired remyelination in MS patients [91,92]. Otherwise, deacetylation is predominant in early MS lesions, suggesting that a disbalance between the acetylation and deacetylation patterns could be considered a disease feature [93]. The epigenetic machinery also has a reciprocal regulation related to microRNAs (miRNAs), which are small non-coding RNAs involved in post-translational regulation [94]. The altered expression and total levels of the survivin proteins related to cell apoptosis have been described as potentially regulated by miRNAs, resulting in auto-reactive lymphocytes [95]. In RRMS patients, a two-fold upregulation of the mRNA expression of survivin was found in CD4^+^ T cells, displaying a positive correlation with the dysregulation of the miRNAs miR-485 and miR-708, which play significant roles in the apoptosis of CD4^+^ T cells [95,96]. 

Dysregulation of these miRNAs can affect immune response by affecting their role in the signal cascades related to the differentiation of T CD4^+^ cells [97]. Upon EAE development, miRNAs were found to be altered; therefore, this parameter could be used as a biomarker for disease activity in MS [98,99]. Its utility has also been described in clinical studies with pediatric multiple sclerosis patients, where miRNA and target genes were used to predict the phenotypes associated with MS pathology, which could also be related to another pathway involving thyroid hormone signaling, thyroid cancer, immunological functions, cell cycle, autophagy-related processes, and ATPase activity [100]. Nevertheless, these alterations in the epigenetic pathways could be essential in studying new therapies [77]. Epigenetic changes are also related to MS through the environmental aspects mentioned before; for example, it has been described that air pollution composed of diverse compounds can induce epigenetic changes and reduce DNA integrity by impairing brain chromatin silencing [101,102]. Further, smoking habits increase MS risk [103] by affecting DNA methylation [104].

## 6. Risk Factors in Postnatal Life Related to MS Pathogenesis

The etiology of MS is complex, and growing evidence has shown the impact of nutritional factors, obesity, hormones, and antioxidant capacity and that various environmental and physiological factors, such as smoking tobacco, vitamin D levels, and the microbiota composition, are related to MS intensity. Therefore, here, we discuss how these factors are related to dysregulation of the immune components that are strongly associated with the development of MS.

### 6.1. Conditions of Birth and Newborn Feeding

For instance, an observational study found that maternal illness during pregnancy led to a 2.3-fold increase in the development of MS in the offspring, while cesarean delivery acts as a protective factor with a 60% reduction in MS risk [105]. At the same time, maternal illness, cesarean delivery, birth weight, and socioeconomic status influence the risk of pediatric onset of MS [105,106]. Moreover, it was observed that male patients carrying the HLA-DRB1*15:01 polymorphism who were breastfed for four months or longer have a reduced risk of MS development [107,108]. In fact, breastfeeding provides bidirectional benefits by lowering the risk of developing multiple sclerosis in both the child and the mother [109]. Furthermore, it contributes to the maturation of the gut barrier [106]. In addition, breastfeeding has immunomodulatory effects, helping the maturation of the neonatal immune system by the proliferation of Treg cells and reducing pro-inflammatory cytokines’ production [110]. In this context, it is important to highlight the protection that is intrinsically delivered by breastfeeding. Breastfeeding favors the proper maturation of the immune system and substantially decreases the predisposition of the offspring to develop inflammatory diseases such as MS.

### 6.2. Reduced Antioxidant Capacity

Activated microglia and infiltrated macrophages produce molecules, such as superoxide, hydroxyl radicals, hydrogen peroxide, and nitric oxygen, which in large amounts contribute to the risk of MS lesions’ progression [111] by a fast activation of the microglia and astrocytes when oxidative stress is related to an imbalance between free radical production and the antioxidant defensive system [112]. There is a risk that the decrease in the total antioxidant capacity is related to an enhanced proinflammatory status [113] and disease development, particularly in the early stages of MS [114,115]. Reactive oxygen species are needed for T CD4^+^ cell activation through metabolic reprogramming to minimize aberrant immune responses in autoimmunity [116]. These observations are thought to be related to the differences in the mitochondrial activity that were observed in MS patients’ lymphocytes [117]. MS patients’ periphery T cells present an increased superoxide anion production together with a significant decrease in the antioxidant capacity and an alteration in mitochondrial protein expression. This suggests a mitochondrial function impairment in these cells that contributes to MS pathogenesis, promoting chronic oxidative stress that leads to the production of reactive oxygen species, i.e., by the NADPH oxidase 2 (Nox2), a redox scavenger enzyme, worsening EAE pathogenesis [118]. Therefore, lower lymphocyte mitochondrial activity in these patients has been associated with MS severity [117]. This evidence leads us to consider that oxidative stress can lead to an imbalance in the metabolic state of MS patients; this aspect is addressed by *Bhargava* et al. (2017), who found impaired redox homeostasis and a glutathione metabolism with changes in the γ-glutamyl amino acids, as well as alterations in various amino acids, such as branched-chain amino acids (i.e., alanine, arginine) and phenylalanine and lysine metabolites. They also found several xenobiotic metabolites and compounds that reflect an altered metabolism of gut microbiota. Metabolic alterations in the redox homeostasis after vitamin D supplementation have also been shown, in which the levels were associated as one of the risk factors for developing MS [119].

### 6.3. Vitamin D Intake in Adulthood

The status of vitamin D (1,25(OH)_2_D3) has been well documented as a genetic and environmental factor involved in MS pathogenesis [120]. It has been described that vitamin D has immune modulating potential and is protective against MS development, since daily intake has been associated with reduced disease risk with sun exposure and vitamin D [121]. Elevated vitamin D blood levels have also been associated with a lower risk factor of MS [122]. In the T cell population of RRMS patients, vitamin D promotes differentiation from Th1 to a Tr1 by crosstalk with CD46 and influences immune modulation by modifying the phenotype of immune cells in patients supplemented with vitamin D [123]. Moreover, CD46 co-stimulated T cells had increased vitamin D receptor gene (*VRD*) expression, as described in other autoimmune diseases, thereby influencing the role of immune function [124]. This observation has been related to the increased methylation observed in the VRD receptor gene of T cells from RRMS patients’ peripheral blood, suggesting an important role for vitamin D [125]. Studies related to the administration of vitamin D_3_ to MS patients have reduced the mRNA expression levels of the proinflammatory cytokines IL-17A and IL-6 and increased the mRNA expression levels of the anti-inflammatory interleukin IL-10 [126]. The suppression of UV light has been related to an increase in the risk of MS given a vitamin D production impairment. Furthermore, it has been shown in vitamin D-deficient mice induced with EAE that even without vitamin D, but with its receptor expressed, UV light exposure can protect them from EAE; however, the cellular mechanisms are still unknown [127].

### 6.4. Intestinal Microbiota

It has been described that low exposure to pathogens in early life could be a risk factor for MS development [128,129]. This is related to the education of the immune system and can be achieved by viruses, parasites, and pathogenic bacteria, which provide the protective effects on autoimmunity. Likewise, the modulation provided by the gut microbiota and the regulation of innate immunity by TLR2’s tolerance for microbial products are deficient in MS [130,131]. Dysbiosis of gut microbiota is a risk factor for MS development, as it has been found in fecal microbiome analyses of MS patients. Their distinct microbial profile is characterized by a lower abundance of Bacteroidetes, Veillonellaceae and an increased quantity of bacterial families such as Ruminococcaceae, Clostridiales, Desulfovibrionaceae [132,133]. Using mice models of EAE that have been constipated to provoke artificial alteration in the gut microbiota composition leads to an exacerbated EAE. These changes in the abundance and diversity of gut microbiota increase intestinal and BBB permeability, aggravating the severity of EAE. [134]. This aspect has also been found in MS patients with increased circulating tight junction proteins, such as occludin and ZO-1 protein levels, which are interpreted as the intestinal barrier and/or BBB being damaged [135]. Bacteria products, such as short-chain fatty acid (SCFA), have been related to MS pathogenesis [136]. These are bioactive metabolites that present immunomodulatory actions and products of the fermentation of dietary fiber, and 95% of the total is composed of acetate, propionate, and butyrate [137,138]. In vitro studies showed that SCFAs like propionate (PA) can promote polarization of naïve CD4^+^ T cells into Tregs [139]. Administration of PA to a mice model of EAE showed an increased content of IL-10 together with a Treg cells’ frequency showing a beneficial effect of PA on control of EAE symptoms [140], suggesting a key role for PA in restoring Treg/Teff cells balance, which has also been observed in MS patients [141,142,143]. Propionate has been suggested to have a regulatory role with regard to immune cells [144]. During the development of an early form of an MS-like disease known as a clinically isolated syndrome (CIS), patients presented low levels of propionate [141]. Low serum levels of propionate have been implicated in changes related to the function of germinal centers’ frequencies of circulating follicular T cells, follicular helper cells CD4^+^CXCR5^+^, T follicular regulatory cells and CD4^+^Foxp3^+^CXCR5, and IL-10-producing B cells. Hence, propionate could be potentially important in counteracting MS pathogenesis due to its modulation on the immune system [141]. There are some contradictory results regarding acetate levels, given that high levels of acetate in the plasma of MS patients is correlated with an expanded disability status scale and increased IL-17^+^ T cells [142]. On the other hand, Olsson et al. (2021) demonstrated that low acetate levels in MS patients negatively correlate with IFN-γ levels, while high butyrate levels positively correlate with pro-inflammatory cytokine levels [135]. 

### 6.5. Nutritional Factors Involved in MS Development

Diet and nutritional status have been suggested as plausible factors regarding MS development and pathogenesis [145]. Epidemiological studies have proposed that exacerbated consumption of saturated fat from animal origins influence the MS development [145]. Likewise, alcoholic beverages, sweets, smoke products, coffee, and tea have been defined as products related to the development of MS; however, how they could be involved in MS physiopathology is not completely clear [146]. Interestingly, diets with high gluten and milk are common in areas with increased prevalence of MS; in particular, Europe and North America [147]. On the other hand, malnutrition is another factor related to MS but not to its development. In fact, malnutrition exacerbates existing symptoms of MS, such as muscle wasting, weakness, and fatigue [148]. Moreover, it has been shown that eating disorders are correlated with an increase in somatic diseases associated with autoimmune or autoinflammatory etiology [149,150,151]. Raevuori et al. (2014), evaluated the risk of autoimmune diseases in individuals with a previous diagnosis of eating disorders, finding an incidence of 0.3% for MS [152]. This evidence supports the need for research about the nutritional status of the worldwide population, given that these disorders could increase the predisposition to develop MS.

## 7. Risk Factors in Pregnancy for the Offspring to Suffer MS in Adult Life

Evidence in the literature proposes that maternal health during pregnancy will impact the development of the immune system, and it could have consequences for the onset and intensity of MS in the adulthood. Therefore, in this section, important aspects of the immune system’s development that relate to risk factors in gestation for MS in adulthood will be mentioned. 

### 7.1. Adaptative Immune System Development during Gestation

The onset of hematopoiesis, stem cell migration, and cell expansion and colonization of bone marrow and the thymus are the first three phases of immunological development that begin during gestation (Figure 2) [153,154]. The first blood cells are extraembryonic and develop in close association with endothelial cells of the yolk sac; shortly after, the liver and fetal bone marrow (BM) are subsequently populated by progenitors derived from the yolk sac and hematopoietic stem cells (HSC) derived from the mesoderm layer of the embryo [155,156]. Fetal liver hematopoiesis is detectable at approximately the sixth week of gestation. To a lesser degree, the liver and the spleen carry the burden of hematopoiesis until shortly after birth [155]. The thymus is the organ that provides the environment for T cell development [156]. Lymphoid progenitors originating in the fetal liver migrate to the thymus, where they develop in naive T cells during the first trimester of fetal development [153]. The development and maturation of the thymus are mediated by an interaction between thymic stromal cells and immune compartments, including thymic epithelial cells, mesenchymal cells, and early developing thymic progenitors that promote the development and maturation of T and other immune cells [156]. HSC-like progenitors, macrophages, mast cells (MC), natural killer (NK) cell progenitors, and innate lymphoid cell progenitors (ILC), together with megakaryocytes and erythroid cells, have been detected during the first gestational trimester [156]. Mature neutrophils are present at the end of the first trimester [153]. Lymphocyte maturation and the development of central tolerance in the thymus occur within the second gestational trimester [157]. Regulatory T cells (Tregs) that participate in the active tolerance process increase their population between the first and second trimesters of pregnancy, generating a pool of fetal Tregs [157] (Figure 2). B cell precursors are present in the fetal liver in the first weeks of gestation, and these cells will give rise to mature B cells after the first trimester [156]. BM becomes the primary source of B lymphocytes during the second trimester; mature B lymphocytes then colonize the spleen, forming a diverse repertoire of B lymphocytes in preparation for antigen exposure after birth [156]. Importantly, adequate maternal nutrition is necessary to develop fetal and neonatal immune responses and immune cell proliferation [157]. Maternal deficiency in certain micronutrients has been reported to affect the immune system, as folate is necessary to maintain the Tregs; vitamins A and D, which are essential to cell-mediated and humoral immune responses; iodine, for production of thyroid hormones; and zinc for lymphocyte activity [157]. Consequently, maternal T_4_ deficiency during the first trimester of pregnancy may affects the development of the fetal immune system. Assuming this occurs during early pregnancy, such deficit could impairs the outcome of the immune cells of the progeny, leading to an increased susceptibility to autoimmune diseases.

### 7.2. Vitamin D

The reduced intake of vitamin D, which has an impact on the predisposition to suffer MS, has also been described during pregnancy. Indeed, maternal vitamin D deficiency is a predisposal risk factor for MS development in pregnant women and their offspring [158,159,160]. Vitamin D modulates the immune system by promoting functional changes, preventing inflammation, and protecting maternal and fetal health [159]. It has been demonstrated that the fetus depends entirely on the mother’s supply of vitamin D during the gestational period [161]. Munger et al. (2017) showed that children born from women deficient in vitamin D in early pregnancy had a 90% increased risk of developing MS in adulthood. Moreover, it was highlighted that when stratifying by the sex of the child, an association was established in female children [158]. On the other hand, increasing exposure to vitamin D during intrauterine life may reduce the risk of MS in the offspring [162]. A retrospective study showed that the relative risk (RR) of MS was lower among women born from mothers with a high vitamin D intake during pregnancy [162]. Hence, the intake of adequate vitamin D levels during pregnancy significantly reduces the risk of the offspring to develop autoimmune diseases, such as MS, particularly in the female progeny.

### 7.3. Maternal Glucocorticoids Related to Fetal Development

Glucocorticoids (GCs) are essential steroid hormones for daily functioning in mammals. They are related to a several processes, such as metabolism, water and electrolyte balance, growth, cardiovascular function, immune response, reproduction, cognitive functions, and development [163]. GCs are mainly synthesized in the adrenal gland cortex, along with aldosterone and dehydro-epi-androsterone (DHEA), which are the precursors of testosterone and estrogen. Adrenal GCs’ production is regulated by the hypothalamic–pituitary–adrenal (HPA) axis [164]. Under basal or unstressed conditions, GCs are released into the bloodstream in a circadian rhythm characterized by peak levels during the active phase (the morning in humans and at the beginning of nighttime in nocturnal animals such as mice). On the other hand, under physiological conditions such as an immune response or emotional stress, the HPA axis increases its activity, releasing the corticotropin-releasing hormone (CRH) and arginine vasopressin (AVP) (from the hypothalamic paraventricular nucleus (PVN)), which binds to CRH-R1 and V1B receptors, respectively, located in the anterior pituitary. This induces release of the adrenocorticotrophic hormone (ACTH) into circulation. ACTH will stimulate the adrenal gland to synthesize and secrete GCs’ hormones (cortisol) into circulation. In turn, the HPA axis is subject to negative feedback inhibition by GCs [164]. Once in the bloodstream, GCs are transported, bound to plasma proteins, which keep the GCs inactive. The corticosteroid-binding globulin (CBG) is the primary GC-binding protein in the plasma, with 90% of the GCs bound to it [165]. GCs diffuse through the cell membrane to enter the cells; nonetheless, the availability of GCs in the cytosol is regulated by the balance of active and inactive forms. Inactive cortisone, in humans, or 11-dehydrocorticosterone, in mice, is activated into active cortisol, in humans, and corticosterone, in mice, through an enzymatic reaction catalyzed by 11β-hydroxysteroid dehydrogenase 1 (11β-HSD1). In contrast, the 11β-HSD2 performs the opposite reaction [166]. Active GCs bind to their receptor in the cytoplasm (called glucocorticoid receptor or GR), a protein of 97kDa that belongs to the nuclear receptor superfamily of transcription factors (TFs) and which is constitutively expressed in the human body; nevertheless, there are different isoforms of this protein that exert different cellular and tissue-specific effects [167].

In general, maternal hormones are essential for proper fetus development. In this context, GCs in pregnancy are relevant for the fetus and the mother. It has been detected that GCs have a dramatic increase of up to 20-fold in the middle of pregnancy, participating in satisfying the increasing energy demands, prenatal programming of the offspring phenotype, and modulating the maturation of fetal organs [168,169]. This extreme rise in the circulating GCs is due to the estrogen stimulation of corticosteroid-binding globulin with an increase in bioavailable cortisol levels [170]. In addition, the placenta secretes large amounts of CRH into the bloodstream, which stimulates the maternal pituitary gland, increasing ACTH and, consequently, cortisol levels. Then, maternal cortisol stimulates placental CRH synthesis, performing a positive loop driven to maintain high cortisol levels in pregnancy [171]. Hence, the maternal HPA axis undergoes dramatic changes during pregnancy, particularly its increased function in this period. Although the GCs are relevant for development, overexposure of the developing fetus to GCs is also related to disease later in life. Therefore, the fetus is protected from high cortisol by the activity of the 11β-HSD2 enzyme [172].

GCs exert immunoregulatory and suppressive functions by pleiotropic effects on the immune system [173]. In pregnancy, GCs are involved in regulating aberrant immune responses in the maternal–fetal interface and are helpful in protecting the fetus from possible rejection. On the other hand, maternal GCs are related to fetal immune development [174]. GCs favor liver and bone marrow erythropoiesis and myeloid hematopoiesis, promoting hematopoietic stem cell (HSC) differentiation to common myeloid progenitors (CMP) and common lymphoid progenitors (CLP). In addition, GCs directly affect bone marrow stromal cells that, through the soluble factors’ secretion, stimulate HSC migration, proliferation, and differentiation steps [175]. Furthermore, endogenous GCs regulate the double-positive (DP) thymocyte maturation. Finally, HPA axis programming is pivotal for the postnatal immune response; in fact, increased levels of CRH and AVP potentiate altered innate and adaptive immune responses in the offspring, i.e., altered monocyte, macrophages, and dendritic cell tolerance towards pathogens or excessive mast cell degranulation, which enhances the risk for infection, asthma, and other immune diseases in postnatal life and adulthood [175].

In the context of pregnancy, GCs interact with other pivotal maternal hormones, such as thyroid hormones (THs) [176]. Maternal THs are also present in neuro and immune development and circulating T_4_ derived from the thyroid gland has been seen reaching the adrenal gland. After being converted to T_3_, this biologically active hormone activates its nuclear receptor, which acts as a transcription factor that transactivates expression of the 11β-HSD1 and 11β-HSD2 enzymes [176]. GCs modulate the contents of available THs by influencing deiodinase activities in various tissue targets. On the other hand, in thyroid-related pathologies, such as hypothyroidism, altered GCs’ metabolisms have been observed, with the reduced enzymatic transformation of cortisol to cortisone in humans or 11-dehydrocorticosterone in mice, possibly due to a decreased expression of 11β-HSD2 [177].

### 7.4. Thyroid Hormones in the Development of the Immune System

Development and maturation of the immune system start early in fetal life; any alteration that occurs during the development process has consequences for the offspring [154]. It is widely known that immune functions operate in a coordinated way with other body systems. For example, the circuit between the neuroendocrine and immune systems works bidirectionally: hormones can influence immune cells and immune-derived products affect endocrine and autonomic central mechanisms [178]. B cells’ development and proliferative capacity are suppressed in mice deficient in thyroid hormones, which is why thyroid hormones are required for normal B cell production in the bone marrow through the regulation of pro-B cell proliferation [179]. T and B lymphocyte development is reduced in T3R1- and T3Rα2-deficient mice, which shows that these genes are involved in immune system development, supporting the importance of the crosstalk between the neuroendocrine and immune systems [180,181]. The influence of thyroid hormone deficiency in the development of the immune system has been determined in a zebrafish model, which relates the administration of a methimazole treatment with the suppression of thymus development and a decreased expression of the genes that encode part of the antigen receptors of T and B lymphocytes [182]. The bidirectional communication between the neuroendocrine and immune systems has also been reviewed by Jara et al. (2017) [183]. The mechanisms underlying the modulation of innate immunity at the cellular level mediated by THs are widely reviewed [184]. T_3_ administration increases the respiratory burst activity of isolated PMNLs. In the same way, a decrease in oxidative metabolism has been observed in PMNLs under hypothyroidism [184]. Moreover, under hypothyroidism, the lipid composition of the PMNLs’ membranes is altered, which may be involved in impairing their function [184]. Thyroid hormone deficiency in hypothyroidism-induced neonates has an immunomodulatory effect featured by an increased proportion of splenic CD4^+^ T cells featuring an increased ratio of helper T cells to suppressor/cytotoxic T cells, an increased NK cells’ proportion, and a decreased B cells’ proportion [185]. Hypothyroid conditions can alter immune cell functions, such as chemotaxis, phagocytosis, the generation of reactive oxygen species (ROS), and cytokine synthesis and release in monocytes, macrophages, leukocytes, NK cells, and lymphocytes [186].

### 7.5. Maternal Thyroid Hormones during Pregnancy

The thyroid gland begins its development during the first month of gestation. It comprises thyroid follicles that synthesize and store thyroid hormones (THs) [187]. There are two main THs synthesized by the thyroid gland, L-3,5,3′,5′-tetraiodothyronine, or thyroxine (T_4_), and 3,3′,5-triiodo-L-thyronine, or triiodothyronine (T_3_). T_4_ is synthesized to a greater extent and converted to T_3_ by specific iodothyronine deiodinases [188,189]. T_3_ has been assumed to be the active form of THs and acts as the principal TH mediating metabolic activity [190]. Thyroid hormones are the only iodine-containing compounds with an established physiological significance in vertebrates. Iodine is an indispensable component of the thyroid hormones; ingested iodine is absorbed through the small intestine and transported from the plasma to the thyroid gland [191]. THs exert genomic and non-genomic effects. To carry out the genomic effects, THs enter target cells through specific transporters [188,192]. The intracellular concentration of T_3_ is determined by the activity of deiodinases D1 and D2 that convert T_4_ into T_3_. Then, T_3_ enters the nucleus and binds to nuclear receptors of thyroid hormone (TRs) to regulate the expression of the T_3_ target genes [188,192]. In general, the action of TRs depends on interaction with DNA sequences, called thyroid hormone response elements (TREs), present in the regulatory regions of the TH target genes and formed by the AGGTCA hexamer consensus motif [193]. Non-genomic effects are carried out through interaction with the integrins present in the plasma membrane, which mediate the intracellular signaling pathways that depend on the activation of phospholipase C (PLC), protein kinase C (PKC), mitogen-activated protein kinase (MAPK 1 and 2); thereby, the non-genomic actions of thyroid hormone contribute to the structure of cells and their basal metabolic rate and can regulate the proliferation of cells [194]. During pregnancy, demands on maternal THs’ production significantly increase by 20–50% to maintain a euthyroid state [195]. Maternal T_4_ is essential in the first half of pregnancy; transplacental passage of the maternal thyroid hormone occurs before the fetal thyroid gland begins to function and provides significant hormone support to the fetus [195,196]. T_4_ is the main TH transferred across the placenta and enters the brain more readily than T_3_ [197]. In the brain, two of the enzymes, iodothyronine deiodinase (D2 and D3), are responsible for maintaining the availability of T_3_ to fulfill its biological functions [198]. Fetal programming is defined as a critical period of development during embryonic and fetal life in which the tissues and organs are created, and the associated functions are subject to a configuration that determines the physiological and metabolic responses that will be carried out in adulthood; thus, alterations during these periods that lead to structural and functional changes, or any other alteration, can have permanent undesirable consequences [199]. The development of fetal organs, specifically the brain, is susceptible to external disturbances, such as alterations in the supply of thyroid hormones, that can have consequences on the anatomy and brain function of the offspring. Maternal thyroid dysfunction during pregnancy may impair the cognitive and motor developmental abilities of the offspring [200]. Among the main side effects of TH deficiency during the first trimester of gestation on CNS development, a reduced expansion of neural progenitors, impaired neuronal migration, and decreased expression of factors involved in neuronal differentiation have been described [201]. During pregnancy, women are more susceptible to hormonal alterations [202]; the environment is an important factor that can affect the endocrine status, especially in promoting autoimmune diseases, such as autoimmune thyroid diseases [203]. Although little is known about the relationship between thyroid disorders and multiple sclerosis risk, the importance of educating patients with relapsing–remitting multiple sclerosis about thyroid dysfunction has been described [204].

### 7.6. Gestational Hypothyroidism and Hypothyroxinemia Increase Susceptibility to Suffering MS

Alterations of thyroid function, such as hypothyroidism and hypothyroxinemia, are common in pregnant women [198,205]. Hypothyroidism is a condition that is characterized by a low concentration of THs [206]. Hypothyroidism is one of the most common maternal thyroid dysfunctions during pregnancy [205]. The prevalence of gestational hypothyroidism is estimated to be 0.3–1.9% for overt hypothyroidism and 1.5–5% for subclinical hypothyroidism [207]. Several studies have demonstrated the detrimental consequences of gestational hypothyroidism in offspring, with a broad spectrum of alterations. Gestational hypothyroidism alters the cardiovascular system of the offspring, increasing the recording of systolic and diastolic pressure values and decreasing the heart rate [208]. Moreover, gestational hypothyroidism can influence the outcome of the CNS to an inflammatory disease, increasing the severity in adult offspring [209]. Albornoz et al. (2013) [209] showed that TH deficiency during gestation significantly increased the severity of EAE in the female offspring. Here, the adult offspring gestated under hypothyroid conditions suffers a significant increase in spinal cord plaques demyelination and immune cell infiltration, together with a high percentage of oligodendrocytes death in spinal cord sections. These results are the first to relate TH deficiency during pregnancy with severe inflammation of the CNS of the offspring (Figure 3).

There is a relationship between gestational hypothyroidism and MS due to the THs’ effect on the neural organization and synaptogenesis, neonatal neuroendocrine system, embryo development, and fetal growth [210]. On the other hand, hypothyroxinemia (HTX) is an asymptomatic condition for the mother characterized by a low serum concentration of free thyroxine (fT_4_) in conjunction with a normal thyrotropin or thyroid-stimulating hormone (TSH) [198,211]. HTX is highly frequent during pregnancy (1.3%), with notably detrimental consequences for the fetus [207]. The evidence found in the literature supports the notion that hypothyroxinemia is 100–200 times more frequent than congenital hypothyroidism [212]. Although the causes of maternal HTX have not been entirely determined, it has been established that HTX in pregnancy could be due to the combination of an increased maternal thyroid stimulation to supply the high demand for THs, a dietary iodine deficiency, and an increased renal iodine clearance [198,207,213]. Gestational HTX imprints a reduced suppressive capacity in the offspring, and this could enhance the severity of an inflammatory challenge, such as EAE [214]. Haensgen et al. (2018) [214] evaluated the effects of gestational HTX on the offspring. In this work, a decreased suppressive capacity of Treg cells contributes to an early and more severe EAE. Furthermore, in vitro experiments has demonstrated that splenic CD4^+^ T cells from the offspring gestated in HTX have a reduced capacity to differentiate into Treg’s phenotype, accompanied by a decreased ability to secrete IL-10 [214]. Moreover, in vitro assays of Treg cells from mice gestated in HTX, showed a reduced capacity to suppress the proliferation of Teff cells [214]. Control mice that were adoptively transferred with Treg cells from mice gestated in HTX and induced with EAE suffered a more intense EAE compared to those that received Treg cells from mice gestated in euthyroidism [214]. These findings highlight that a T_4_ deficiency during gestation decreases the regulatory capacity of the Treg cells of the offspring.

## 8. Conclusions

The evidence obtained from the human and animal models of MS supports the notion that the causes and the outcome of this autoimmune disease are highly complex, and several factors are involved. Clearly, there is an essential balance between tolerance and immunity and/or Treg and Th17 lymphocytes for the outcome of MS; however, other factors could help or worsen it, such as nutrition, microbiota, BBB permeability, and the autoreactivity of T cells. In this review, we highlighted the role of gestation and gestational factors in the MS susceptibility of the offspring. We gave importance to maternal TH deficiency during pregnancy, such as those eliciting hypothyroidism and HTX, because these conditions affect the development of the CNS [200], reduce oligodendrocyte differentiation, increase the inflammatory response of astrocytes and microglia [209], and affect the immune system’s development. Moreover, both gestational HTX and gestational hypothyroidism increase the pro-inflammatory immune response in EAE [209,214] and infection [215]. Regretfully, studies relating TH deficiency during gestation with an increased susceptibility to MS in the offspring were derived from animal models due to the complexity of obtaining retrospectively data from humans gestated under TH deficiency. Specially, gestation in HTX is a delicate issue given that this condition is not diagnosed and, consequently, is not treated in pregnant women. In summary, humans are exposed to several risk factors during gestation and postnatal life that will increase their susceptibility to suffer MS. Special emphasis is given to the evidence that supports the consequences of maternal thyroid hormone deficiency on the progeny to suffer more intense autoimmune disorders, such as MS. Therefore, robust clinical and epidemiological studies are highly recommended because they contribute to elucidate the impact of gestational factors, such as maternal TH on the fetal development and their influence on the offspring predisposition to suffer autoimmune diseases in adulthood.

## Figures and Tables

**Figure 1 ijms-23-07080-f001:**
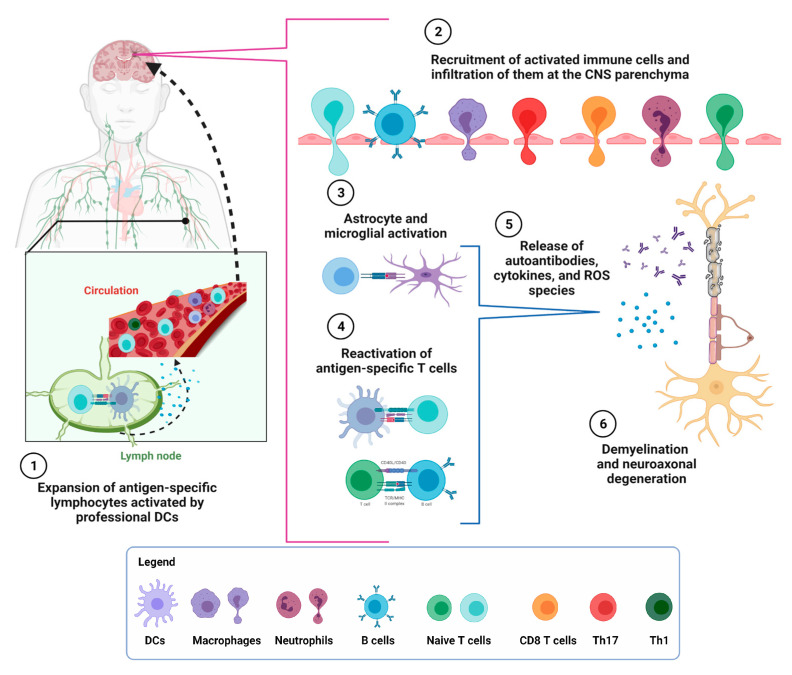
The immune interplay in MS pathogenesis. MS has a heterogeneous presentation and a robust immune component that underlies its development. (1) Autoantigens against the myelin basic protein (MBP) and the myelin oligodendrocyte glycoprotein (MOG) are released into circulation and captured by professional dendritic cells (DCs), which present them to naive T cells in secondary lymphatic organs. In parallel, B cells are activated mainly in CSF. (2) Anti-MBP and anti-MOG antibodies and chemokines allow the recruitment of activated T effector and innate cells infiltrating the CNS parenchyma due to altered permeability of the blood-–brain barrier (BBB). (3–4) Effector T and B cells are re-activated inside the CNS by astrocyte and microglial cells, (5) increasing local pro-inflammatory cytokines and reactive oxygen species (ROS) production, (6) which destroys the myelin sheath, thereby causing demyelination, oligodendrocyte loss, gliosis, and neuroaxonal degeneration. This damage is observed in patients as demyelinated areas called plaques, which are the distinctive feature of MS. Created with BioRender.com (accessed on 1 June 2022).

**Figure 2 ijms-23-07080-f002:**
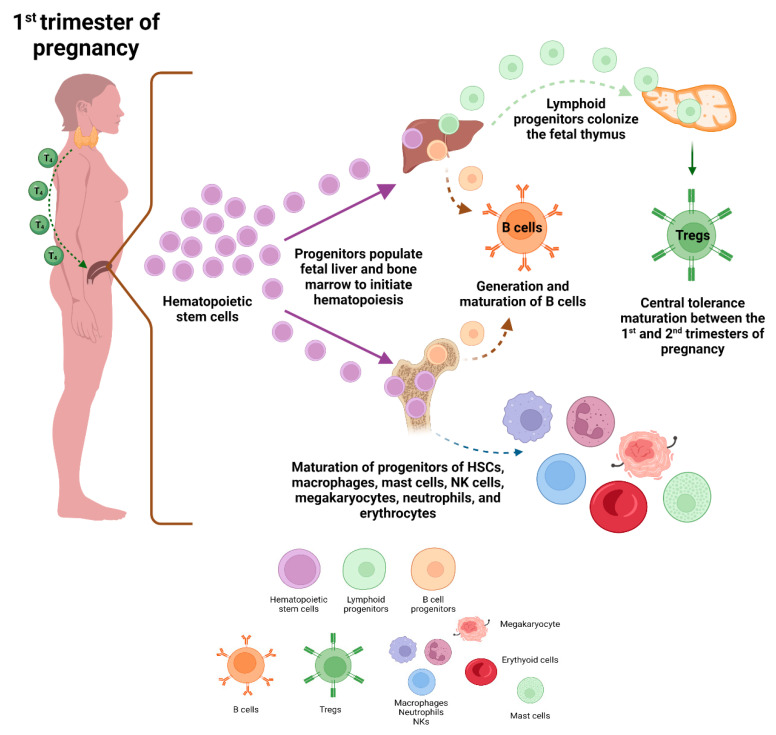
Fetal immune development. Placental hematopoietic stem cells populate the liver and bone marrow of the fetus, where hematopoiesis takes place. Lymphoid progenitors migrate and colonize the fetal thymus, where lymphocyte maturation and central tolerance occur. In parallel, hepatic progenitors also mature to form B cells during the first trimester. On the other hand, bone marrow progenitors will mainly develop into B cells, macrophages, mast cells, NK cells, neutrophils, megakaryocytes, and erythroid cells. As maternal thyroid hormones are pivotal for immune development during the first trimester, a thyroid deficiency during pregnancy will affect the development of the immune system of the progeny. Created with BioRender.com (accessed on 1 January 2022).

**Figure 3 ijms-23-07080-f003:**
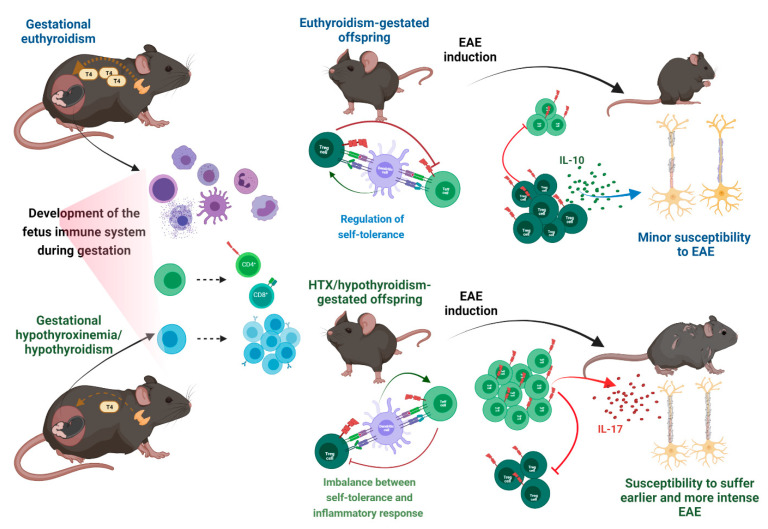
Thyroid hormone deficiency during gestation increases the immune response of the offspring after EAE induction. During the period of gestation, development and migration of immune cells usually occur properly; however, when it takes place under conditions of THs deficiency, this developmental process is altered. The offspring gestated in HTX, or hypothyroidism that was challenged with EAE in adulthood, have an early and severe disease onset that involves a more significant deterioration of the myelin sheath compared to progeny gestated in euthyroidism, which have lower susceptibility to suffering from the disease with less deterioration of the myelin sheath. Furthermore, HTX–Treg lymphocytes have a reduced suppressive capacity and a lower secretion of IL-10, establishing an imbalance of self-tolerance, compared with progeny gestated under euthyroid conditions who have Treg lymphocytes capable of exerting the suppressive effect correctly. Created with BioRender.com (accessed on 1 January 2022).

## Data Availability

Not applicable.

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
