# Peer review of "Risk Factors from Pregnancy to Adulthood in Multiple Sclerosis Outcome"

_ijms, 2022, doi:10.3390/ijms23137080_

Round 1

Reviewer 1 Report

The review tries to connect effect of fetal programming of MS in children by mother hypothyroidism in during pregnancy. This approach is very attractive and necessary for understanding of epigenetic modulation of fetus during pregnancy.

The work describes the problem carefully and to great details. From pathophysiological point of view it could be interesting to complete also explanation how glucocorticoids as leading control hormones in pregnancy should participate in hypothyroidism development. Also immunoendocrine influence of placenta in complicated psycho-neuro-immuno-endocrinal interplay in fetal programming should be mentioned. Other eating disorders, not only obesity, can participate on development of many autoimmune disorders. Is it also a problem of MS?         

Reviewer 2 Report

In this review by Gonzalez-Madrid et al., the aim of the authors is to discuss the relevance of gestational factors that could increase the risk of progeny of developing multiple sclerosis in the adult life.

This kind of review could be of potential interest for this field of study and it could be very relevant for the scientific community, however, some concerns are present that need to be addressed.

A major concern regards the discrepancy between the main objective of the review, as reported in the title of the manuscript, and the main text. Indeed, the whole first part of this review, which describes the innate and adaptive immunological response characterizing the disease and the risk factors in adult life related to its pathogenesis, are not essential and beyond the scope of the manuscript.

Authors should remove this unnecessary description or change the title in order to better describe the content of the whole manuscript.

In addition, the paragraph describing gestational risk factors for developing MS in adult life is essentially confined to the role of thyroid hormones and omits the description of contribution all other possible risk factors that are only listed.

The role of maternal vitamin D levels during pregnancy as a risk factor for the development of MS in offspring is only mentioned in section 4.4. This aspect, for example, should be more stressed, described in more details and included among the gestational risk factors for MS developing (paragraph 5).
